# PHYSICS-INFORMED DECENTRALIZED FEDERATED LEARNING

## ABSTRACT

The integration of domain knowledge into the learning process of artificial intelligence (AI) has received significant attention in the last few years. Most of the approaches proposed so far have focused on centralized machine learning scenarios, with less emphasis on how domain knowledge can be effectively integrated in decentralized settings. In this paper, we address this gap by evaluating the effectiveness of domain knowledge integration in distributed settings, specifically in the context of Decentralized Federated Learning (DFL). We propose the Physics-Informed DFL (PIDFL) architecture by integrating domain knowledge expressed as differential equations. We introduce a serverless data aggregation algorithm for PIDFL, prove its convergence, and discuss its computational complexity. We performed comprehensive experiments across various datasets and demonstrated that PIDFL significantly reduces average loss across diverse applications. This highlights the potential of PIDFL and offers a promising avenue for improving decentralized learning through domain knowledge integration.

## 1 INTRODUCTION

Federated Learning (FL) has been introduced as an alternative to classical centralized training to solve different issues including data security, privacy, and data transfer costs, prevalent in distributed environments (AbdulRahman et al., 2020). Indeed, a large number of geographically dispersed devices and sensors are equipped with a local machine learning model nowadays. In FL data are not moved to the central server but stored and analyzed locally, with model parameters shared among nodes. Each node retains its local model, obtained by taking into account the learning process of the whole system. The relationship between global and local models is orchestrated by a central server in *Centralized FL* (CFL), as shown in Figure 1(a), while in *Decentralized Federated Learning* (DFL), there is no centralized aggregator entity, as shown in Figure 1(b). Hybrid solutions, where some nodes operate as aggregator entities, performing parameter analyses and sharing, are considered variations of DFL. A DFL network can be seen as an undirected graph (see Figure 1(b)), where edges represent connections among nodes, and nodes (shown in Figure 1 as the squared box) contain a local dataset and a local model with its parameters, that are shared with other adjacent nodes. DFL addresses issues such as reducing centralized risk, enhancing privacy, optimizing resource utilization, improving scalability, ensuring regulation compliance, and contributing to the democratization of Artificial Intelligence (AI). However, different challenges regarding communication overhead, data distribution, data security, and privacy have been addressed in the literature (for a recent survey on DFL, refer to (Beltrán et al., 2023)).

The utilization of domain knowledge to enhance machine learning performance has been the subject of numerous recent efforts. Systems allowing expressions of domain knowledge through logical formulas are known as *neuro-symbolic* (see (Garcez & Lamb, 2023) as a survey on this topic). The integration of mathematical equations within learning algorithms is studied in the *physics-informed* learning (Karniadakis et al., 2021; Piccialli et al., 2024). A recent research direction following this intuition is the proposal of the Physics-Informed Neural Networks (PINNs), seeking to integrate physics-related domain knowledge, in the form of mathematical equations, as soft constraints into an empirical loss function of a neural network (Krishnapriyan et al., 2021; Hao et al., 2022; Chen et al., 2020). However, the existing studies have taken place in centralized structures, where the dataset and model are under the same administrative authority. Independently of the type of domain knowledge, its use favors the training speed in large-scale datasets and accuracy.

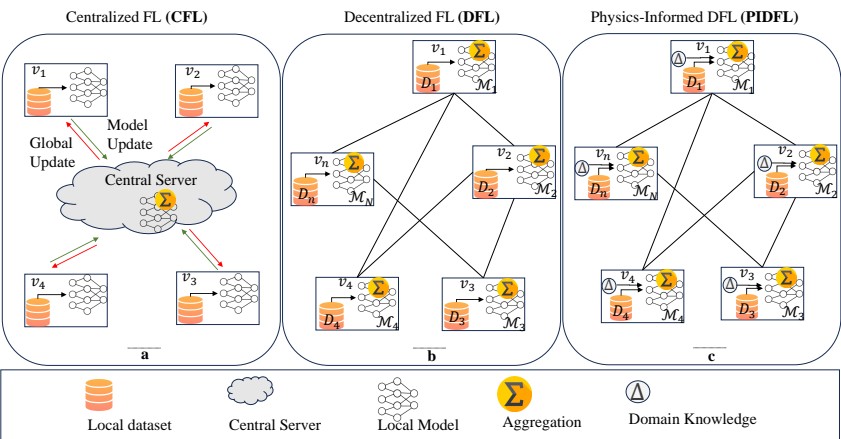

Figure 1: Federated Learning architectures: CFL (a), DFL (b) proposed PIDFL (c).

In this paper, we investigate the possibility of using domain knowledge in *Decentralized Federated Learning (DFL)* framework. Our proposal is motivated by the fact that in DFL the local data can be limited, possibly noisy, and may vary in terms of distribution (e.g. Heterogeneous data) and volume (Beltrán et al., 2023), and thus the use of domain knowledge could improve the learning performance of single nodes, and consequently of the whole DFL system. To this end, we generalize PINN to deal with (decentralized) federated learning and propose an architecture called Physics-Informed Decentralized Federated Learning (PIDFL) (see Figure 1(c)), integrating domain knowledge into decentralized federated learning. While the idea behind the proposed framework is straighforward, we face unique technical considerations including heterogeneous data as it impacts the convergence of the learning process. On the other hand, the decentralized learning setup requires specific adaptation to the standard PINNs due to the practical considerations including the network topology, and communication limitations. Our architecture is suitable for many real-world distributed machine learning applications, especially when dealing with heterogeneous and scarce data during training. To the best of our knowledge, this paper marks the initial attempt to integrate domain knowledge, presented as physical equations, with machine learning in decentralized systems.

**Contributions.** Our main contributions are as follows:

- We propose a general architecture called Physics-Informed Decentralized Federated Learning (PIDFL), that integrates domain knowledge, expressed in terms of differential equations into decentralized federated learning. Our architecture is suitable for many real-world distributed machine learning applications dealing with scarce data during training.

- We propose a data aggregation algorithm for PIDFL called DFLA, prove its convergence, and discuss its computational complexity.

- Performing comprehensive experiments across various datasets, we show that PIDFL significantly improves the performance in terms of average loss. We utilize a non-IID (non-independent and identically distributed) data distribution and compare the performance of the PIDFL in different settings with existing baseline DFL algorithms including well-known Federated Averaging (FedAvg) (McMahan et al., 2017b), Segmented Gossip (SG) (Hu et al., 2019) algorithms.

**Organization.** The rest of this paper is organized as follows. Section 2 recalls the key concepts underlying Partial Differential Equations, Physics-Informed Neural Networks, and Decentralized Federated Learning. Section 3 presents the PIDFL framework, the distributed aggregation algorithm (Section 3.1), and its convergence analysis, computational complexity, theoretical limits, and optimization of hyperparameters. The experimental analysis is presented in Section 4. Related work is discussed in Section 5, before concluding the paper in Section 6.

## 2 PRELIMINARIES

We recall the key concepts underlying Supervised ML, Physics-Informed Neural Networks, and Decentralized Federated Learning.

### 2.1 SUPERVISED MACHINE LEARNING

A supervised (neural network) learning model can be defined as a pair $\mathcal{M} = \langle N, \Theta \rangle$ where $N$ identifies the neural network and $\Theta$ denotes the set of its parameter values. The goal is to build a function $f_N(x; \Theta)$ (or simply $f(x; \Theta)$ whenever the neural network is understood) relating inputs $x$ (also called instances) to outputs $\hat{y} = f(x; \Theta)$ (also called model predictions). The particular relationship between inputs and model predictions is determined by $\mathcal{M}$. To train the model, a loss function $\mathcal{L}(\mathcal{D}, \Theta)$ is adopted over a training dataset $\mathcal{D}$ consisting in pairs $(x, y)$, where $x$ is an instance and $y$ is its corresponding label (also called ground truth). The loss function quantifies the mismatch between the model prediction $\hat{y} = f(x; \Theta)$ and the ground truth $y$ over all pairs $(x, y)$ in $\mathcal{D}$. Since the function $f(x; \Theta)$ depends on parameters $\Theta$, the goal is to search for the parameter values that minimize the loss. An important neural network learning model is Multi-Layer Perceptron (MLP), that appeared as a building block of several learning architectures (Prince, 2023; Bengio et al., 2017). An MLP $\mathcal{M} = \langle N, \Theta \rangle$, where $N$ has $k$ layers, is defined by a sequence of weighted matrices $\boldsymbol{\omega}^{(1)}, \ldots, \boldsymbol{\omega}^{(k)}$, bias vectors $\mathbf{b}^{(1)}, \ldots, \mathbf{b}^{(k)}$, and fixed activation functions $a^{(1)}, \ldots, a^{(k)}$. [1] Given an input instance $x$, we inductively define $\mathbf{h}^{(i)} = a^{(i)}(\mathbf{h}^{(i-1)}\boldsymbol{\omega}^{(i)} + \mathbf{b}^{(i)})$ with $i \in \{1, \ldots, k\}$, assuming that $\mathbf{h}^{(0)} = x$. The output of $\mathcal{M}$ on $x$ is defined as $\mathbf{h}^{(k)}$.

### 2.2 PARTIAL DIFFERENTIAL EQUATIONS AND PHYSICS-INFORMED NEURAL NETWORKS

Partial differential equations (PDEs) are typically derived from fundamental governing principles such as the conservation of mass or energy, these PDEs often lack exact analytical solutions in many real-world scenarios. The following abstraction captures many of the issues associated with a PDE constraint (Krishnapriyan et al., 2021; Moin, 2010):

$$\mathcal{F}(c(x_1, \ldots, x_n)) = 0, \quad \text{with} \ [x_1, \ldots, x_{n-1}] \in \Omega, \quad x_n \in [0, H]$$

where $\mathcal{F}$ is a differential operator representing the PDE, $c(x_1, \ldots, x_n)$ is the state variable (i.e., the parameter of interest), $x_1, \ldots, x_{n-1}$ denote space, $x_n$ denotes the time, $H$ is the time horizon, and $\Omega$ is the spatial domain. Since $\mathcal{F}$ is a differential operator, in general one must specify appropriate boundary and/or initial conditions to ensure the existence/uniqueness of a solution.

**Example 1.** *Considering a pollutant's dispersion scenario, $c(x_1, x_2, x_3)$ represents the pollutant concentration at time $x_3$ at the coordinates of $(x_1, x_2)$. Moreover, the pollutant's dispersion could be modeled by Advection-diffusion (Lanser & Verwer, 1999) with the following differential equation:*

$$\Delta: \ \delta\left(\frac{\mathrm{d}^2 c}{\mathrm{d}x_1^2} + \frac{\mathrm{d}^2 c}{\mathrm{d}x_2^2}\right) - \frac{\mathrm{d}c}{\mathrm{d}x_3} - \left(\rho_1 \frac{\mathrm{d}c}{\mathrm{d}x_1} + \rho_2 \frac{\mathrm{d}c}{\mathrm{d}x_2}\right) + \sigma = 0 \tag{1}$$

*where $\rho_1$ and $\rho_2$ are wind velocity components, $\delta$ is the so-called diffusion coefficient, and $\sigma$ represents the source of pollutant.* □

Current research on PINN aims to integrate partial differential equation as soft constraints in the neural network's output using an empirical loss function (Hao et al., 2022; Krishnapriyan et al., 2021). The goal is to find the neural network parameters $\Theta$ that minimize $\mathcal{L}(c) + \lambda_{\mathcal{F}} \mathcal{F}(c)$, where $\mathcal{L}(c)$ is the data-fit term (including initial/boundary conditions), and $\lambda_{\mathcal{F}}$ is a regularization parameter that controls the emphasis on the PDE based residual (which we ideally want to be zero).[2] Sharing the same underlying idea, we generalize PINNs within (decentralized) federated learning setting.

---

[1] In MLP, the activation functions are part of the neural network $N$, while matrices $\boldsymbol{\omega}$ and vectors $\mathbf{b}$ constitute parameters $\Theta$.

[2] Loosely speaking, a residual is the error in computing the exact value of $\mathcal{F}(c)$. This is due to the fact that, for many practical use cases, it is not possible to derive closed-form solutions for these problems.

## 2.3 DECENTRALIZED FEDERATED LEARNING

A *Decentralized Federated Learning framework* (or simply DFL) can be intuitively seen as a graph whose nodes can collect and process local data and communicate with the other nodes through edges. More formally, a DFL is a pair $\langle \mathcal{V}, \mathcal{E} \rangle$, where $\mathcal{V}$ is a set of nodes (e.g., agents) and $\mathcal{E} \subseteq \mathcal{V} \times \mathcal{V} \times \mathbb{R}$ is a set of directed edges among pairs of nodes such that there are no two edges $(v_i, v_j, w_{ij})$ and $(v_i, v_j, w'_{ij})$ with $w_{ij} \neq w'_{ij}$. An edge $(v_i, v_j, w_{ij}) \in \mathcal{E}$ represents the fact that node $v_j$ receives information from $v_i$ and, as it will be clearer in what follows, the weight $w_{ij}$ (with $w_{ij} \geq 0$) intuitively represents the importance that $v_j$ gives to the received information. Each node $v_i \in \mathcal{V}$ balances the information received from its neighbors with its local information—to this end we also assume the existence of 'self-loop' edges $(v_i, v_i, w_{ii}) \in \mathcal{E}$. Moreover, in DFL it is also assumed that the communication is symmetric, that is $(v_i, v_j, w_{ij}) \in \mathcal{E}$ if and only if $(v_j, v_i, w_{ji}) \in \mathcal{E}$, although $w_{ij}$ and $w_{ji}$ may differ. Each node $v_i \in \mathcal{V}$ contains a local dataset $\mathcal{D}_i$ and a local model $\mathcal{M}_i = \langle N_i, \Theta_i \rangle$ parameterized by $\Theta_i$. It is also assumed that nodes share the same neural network, that is $N_i = N_j$ for any pair of nodes $(v_i, v_j)$. Thus, we often denote a DFL as a triple $\langle \mathcal{V}, \mathcal{E}, N \rangle$. We use $\mathcal{E}_i = \{ (v_j, w_{ji}) \mid (v_j, v_i, w_{ji}) \in \mathcal{E} \}$ to denote the neighborhood of $v_i$, that is the set of pairs $(v_j, w_{ji})$ where $v_j$ is a neighbor of $v_i$ and $w_{ji}$ is the weight of the edge from $v_j$ to $v_i$ denoting the importance $v_i$ gives to $v_j$. A DFL is said to be *fully connected* if all pairs of nodes are directly connected by an edge, that is the graph $\langle \mathcal{V}, \mathcal{E} \rangle$ is complete.

The training process of a DFL network is delegated to an *aggregation* algorithm, where each node minimizes its local loss by also taking into account the information provided by its neighbors (Beltrán et al., 2023). Most of the aggregation algorithms in the literature share the same underlying (training) idea: each node, at each iteration, update its model parameters by leveraging on its local dataset and the parameters received from its neighbors (Sun et al., 2023; He et al., 2018; Martínez Beltrán et al., 2024; McMahan et al., 2017a). The algorithm ends whenever convergence criteria are satisfied or a maximum number of iterations is reached. This approach of sharing the model parameters instead of raw data is particularly useful for privacy preservation and efficient computation in distributed networks. Thus, DFL proposes many advantages over CFL in terms of privacy preservation, communication efficiency, scalability, and resilience to adversarial attacks.

## 3 PHYSICS-INFORMED DFL FRAMEWORK

In this section we present the *Physics-Informed Decentralized Federated Learning* (PIDFL) Framework (or simply PIDFL).

A PIDFL $\langle \mathcal{V}, \mathcal{E}, N \rangle$ is a specific DFL where each node $v_i$ also contains some physics-related laws, denoted as $\Delta_i$. Motivated by the fact that nodes in DFL typically learn the same phenomena and share the same neural network, we assume that all nodes share the same physics-related laws, that is $\Delta_i = \Delta_j$ for any pair $(v_i, v_j)$ of nodes. Thus, we often denote a PIDFL as a quadruple $\langle \mathcal{V}, \mathcal{E}, N, \Delta \rangle$. For the sake of readability, w.l.o.g. we consider $\Delta$ as a single PDE. It is worth noting that $\Delta$ should be not necessarily applied to all the samples from the local dataset $\mathcal{D}_i$. Thus, we denote with $\mathcal{X}_i \subseteq \mathcal{D}_i$ the subset of $\mathcal{D}_i$ where $\Delta$ is expected to hold. Selection of $\mathcal{X}_i$ depends on the domain-specific knowledge. For instance, in our example (cf. Example 1), air pollutant dispersion can be influenced by wind patterns, urban geometry (e.g., buildings), temperature gradients, and emissions sources. Thus, to ensure that the model learns the initial conditions and source-related terms of the dispersion, the set $\mathcal{X}_i$ might be selected as the data points around known emission sources such as industrial areas, or areas with high levels of human activity. This process can be performed automatically by data-driven or adaptive sampling methods. In data-driven methods, specific patterns or regions within $\mathcal{D}_i$ could be selected while adaptive sampling techniques evaluate the model performance, or the uncertainty in prediction to dynamically identify $\mathcal{X}_i$.

The *PIDFL problem* consists in the individuation of parameters $\Theta_1, \dots, \Theta_n$ that minimize the summation $\sum_{v_i \in \mathcal{V}} \mathcal{L}(\mathcal{D}_i, \mathcal{X}_i, \Theta_i)$, where each local loss function $\mathcal{L}(\mathcal{D}_i, \mathcal{X}_i, \Theta_i)$ is defined as follows:

$$\mathcal{L}(\mathcal{D}_i, \mathcal{X}_i, \Theta_i) = \mathcal{L}^{\mathrm{d}}(\mathcal{D}_i, \Theta_i) + \lambda \mathcal{L}^{\Delta}(\mathcal{X}_i, \Theta_i). \tag{2}$$

In the above equation, $\mathcal{L}^{\mathrm{d}}(\mathcal{D}_i, \Theta_i)$ represents the local loss function based on local data $\mathcal{D}_i$ and local model parameters $\Theta_i$. Moreover, $\mathcal{L}^{\Delta}(\mathcal{X}_i, \Theta_i)$ is the local loss function based on the physics-related law $\Delta$, whereas $\lambda$ is a regularization parameter that balances the data fidelity with the physic-law adherence integrating the latter as a soft constraint.

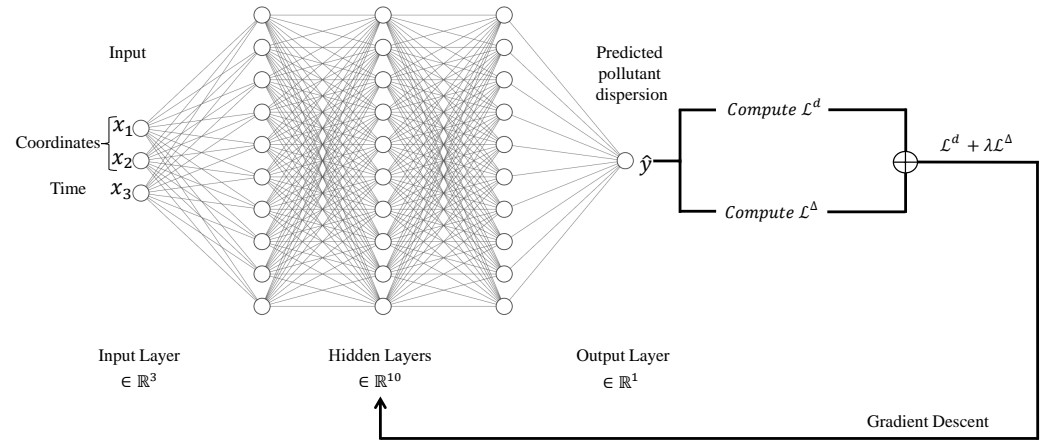

Figure 2: Overview of the training process for local model $\mathcal{M}_i$ contained in any node $v_i$ of a PIDFL framework presented in Example 1.

Let $\hat{y} = f(x; \Theta_i)$ be the model prediction for any element $(x, y) \in \mathcal{D}_i$, $\mathcal{L}^d$ is defined as the mean squared error (MSE) between the prediction and ground truth as follows:

$$\mathcal{L}^d(\mathcal{D}_i, \Theta_i) = \frac{1}{|\mathcal{D}_i|} \sum_{(x,y) \in \mathcal{D}_i} \left(f(x; \Theta_i) - y\right)^2. \tag{3}$$

The physics-informed term $\mathcal{L}^\Delta(\mathcal{X}_i, \Theta_i)$ is formulated based on physics-related law $\Delta$ on a set of data points $\mathcal{X}_i \subseteq \mathcal{D}_i$ as follows:

$$\mathcal{L}^\Delta(\mathcal{X}_i, \Theta_i) = \frac{1}{|\mathcal{X}_i|} \sum_{(x,y) \in \mathcal{X}_i} \left(\texttt{residual}(\Delta, x, \Theta_i)\right)^2 \tag{4}$$

where $\texttt{residual}(\Delta, x, \Theta_i)$ is a function computing the residual at $x$—the larger the value the greater the error; conversely, the smaller the larger the number of data points compatible with $\Delta$.

We next provide the DFLA aggregation algorithm, prove its convergence, and discuss its computational complexity.

## 3.1 AGGREGATION ALGORITHM

We propose an aggregation algorithm for PIDFL called DFLA that runs on each node of the PIDFL network and is used to train multiple local models cooperatively, i.e. they are not trained solely based on individual local data but also consider the learning parameters of its neighbors. Both cooperation and domain knowledge are expected to improve the accuracy of training, especially when local data is scarce. This will be confirmed in our experimental analysis in Section 4.

We now discuss how the proposed distributed algorithm (i.e., Algorithm 1) is performed on (any) node $v_i \in \mathcal{V}$ of the PIDFL network $\langle \mathcal{V}, \mathcal{E}, N, \Delta \rangle$. It takes as input the local data $\mathcal{D}_i$ and $\mathcal{X}_i \subseteq \mathcal{D}_i$, the PDE $\Delta$, the neural network $N$, the set $\mathcal{E}_i$ of pairs $(v_j, w_{ji})$ including both $v_i$'s neighbors and respective importance weights $w_{ji}$, the maximum number of iterations $\tau \in \mathbb{N}$, and the regularization parameter $\lambda$. The algorithm initializes parameters $\Theta_i^0$ (Line 1). Then, at each iteration $t \in [0, \tau - 1]$ it computes the local loss function $\mathcal{L}(\mathcal{D}_i, \mathcal{X}_i, \Theta_i^t)$ as outlined in Eq. 2 (Line 3). Then, node $v_i$ first computes the gradient of the loss w.r.t. the model parameters (Line 4) and then performs an optimizer step for each iteration $t \in [0, \tau - 1]$ (Line 5). That is, it consists of updating (through function $\texttt{update}$) the parameters $\Theta_i$ of local model $\mathcal{M}_i = \langle N, \Theta_i \rangle$ as follows:

$$\widehat{\Theta}_i^t = \Theta_i^t - \mu \nabla(\mathcal{L}^d(\mathcal{D}_i, \Theta_i^t) + \lambda \mathcal{L}^\Delta(\mathcal{X}_i, \Theta_i^t)), \tag{5}$$

---

**Algorithm 1** DFLA($\mathcal{D}_i, \mathcal{X}_i, \Delta, N, \mathcal{E}_i, \tau, \lambda$)

---

**Input:** Local data $\mathcal{D}_i$ and $\mathcal{X}_i \subseteq \mathcal{D}_i$, PDE $\Delta$, neural network $N$, set $\mathcal{E}_i$ of pairs $(v_j, w_{ij})$, maximum number of iterations $\tau$, regularization parameter $\lambda$.
**Output:** Trained model $\mathcal{M}_i = \langle N, \Theta_i \rangle$
 1: Initialize $\Theta_i^0$;
 2: **for** $t \in [0, 1, \ldots, \tau - 1]$
 3:     Let $\mathcal{L}(\mathcal{D}_i, \mathcal{X}_i, \Theta_i^t) = \mathcal{L}^d(\mathcal{D}_i, \Theta_i^t) + \lambda \mathcal{L}^\Delta(\mathcal{X}_i, \Theta_i^t)$;
 4:     $g_\mathcal{L} \leftarrow \nabla \mathcal{L}(\mathcal{D}_i, \mathcal{X}_i, \Theta_i^t)$;                    ▷ *Gradient computation*
 5:     $\widehat{\Theta}_i^t \leftarrow \texttt{update}(\Theta_i^t, g_\mathcal{L})$;                    ▷ *Optimizer step*
 6:     Send $\widehat{\Theta}_i^t$ to neighbors $v_j$ in $\mathcal{E}_i$ and receive $\widehat{\Theta}_j^t$;
 7:     Compute $\Theta_i^{t+1} = \sum\limits_{(v_j, w_{ji}) \in \mathcal{N}_i} w_{ji}\widehat{\Theta}_j^t$;
 8: **return** trained model $\mathcal{M}_i = \langle N, \Theta_i = \Theta_i^\tau \rangle$;

---

where $\nabla$ represents the gradient over the local loss function and $\mu$ is the learning rate that can be explicitly specified or adaptively adjusted by adaptive optimizers (Bengio et al., 2017). Then, at Line 6, node $v_i$ performs the direct Peer-to-Peer communication (Beltrán et al., 2023). Therefore, the model parameters $\widehat{\Theta}_i^t$ are directly sent to their neighbors $v_j \in \mathcal{E}_i$. Since $v_i$ sends and receives updates from their neighbors, the communication is efficient in terms of bandwidth and computational resources. Finally, parameters $\widehat{\Theta}_j^t$ are used to compute $\Theta_i^{t+1}$ at Line 7 as follows:

$$\Theta_i^{t+1} = \sum_{(v_j, w_{ji}) \in \mathcal{E}_i} w_{ji}\widehat{\Theta}_j^t \tag{6}$$

where the weights $w_{ji}$ intuitively represent the importance that node $v_i$, in updating its local parameters $\Theta_i^{t+1}$, gives to the received parameters $\widehat{\Theta}_j^t$. These weights could be uniform or optimized to determine the most influential nodes in the parameter update. After the last iteration $t = \tau - 1$, Algorithm 1 ends returning the trained model $\mathcal{M}_i = \langle N, \Theta_i = \Theta_i^\tau \rangle$ (Line 8).

**Computational Complexity.** The complexity of DFLA is positively related to (*a*) the maximum number of iterations $\tau$, (*b*) the number of neighbors $|\mathcal{E}_i|$, (*c*) the number of data-points $|\mathcal{D}_i|$ and $|\mathcal{X}_i|$, and (*d*) the topology of neural network $N$. Thus, the worst case is whenever $|\mathcal{X}_i| = |\mathcal{D}_i|$ and $|\mathcal{E}_i| = |\mathcal{V}|$. Furthermore, let DFL be the corresponding algorithm in the DFL setting, that is obtained from DFLA by setting $\lambda = 0$. Notably, the overhead caused by the computation of the loss $\mathcal{L}^\Delta(\mathcal{X}_i, \Theta_i^t)$ in DFLA is negligible as its cost is lower than that of computing the gradient during backpropagation (Line 4). Notably, this holds regardless of the neural network $N$; therefore, DFLA and DFL have the same complexity, that is the introduction of the physical law is not a source in complexity.

**Importance Weights.** We now discuss various possible definitions for the importance weights $w_{ij}$. Let $W$ represent the weighted adjacency matrix associated with the graph $\langle \mathcal{N}, \mathcal{E} \rangle$. The choice of $W$ may depend on network topology and communication patterns. However, as it will be clearer in Section 3.2, a significant aspect for achieving better convergence is to ensure that the matrix $W$ is doubly stochastic, i.e. $w_{ij} \geq 0$ and $\sum_j w_{ij} = \sum_i w_{ij} = 1$. For any possible definition of $W$, it is reasonable to set $w_{ij} = 0$ if there is no edge between $v_i$ and $v_j$. In a fully connected network, when there is no prior knowledge about the importance of the nodes, the simplest and most efficient method is to use the uniform distribution, where each node $v_i$ considers the information received from its neighbors to be equally informative, i.e. $w_{ji} = 1/|\mathcal{V}|$ for any node $v_j \in \mathcal{V}$. When equal importance is not desired, to improve the convergence rate, the matrix $W$ can be designed to maximize the *spectral gap*, i.e. the difference between the largest ($\Lambda_1$) and second-largest ($\Lambda_2$) eigenvalue (Vogels et al., 2022). To find the optimal values for the elements in $W$, we need to solve the following optimization problem numerically since there is no closed-form solution in general. [3]

$$\max_W (1 - \Lambda_2(W))$$

$$\text{subject to} \quad W\mathbf{1} = \mathbf{1}, \quad W^T\mathbf{1} = \mathbf{1}, \quad w_{ij} \geq 0, \quad \forall i, j.$$

---

[3]Recall that, whenever $W$ is doubly stochastic, $\Lambda_1 = 1$ and thus the spectral gap can be defined as $1 - \Lambda_2$.

It is worth noting that doubly stochastic property on $W$ is easily met on fully connected frameworks, while for partially connected frameworks it is not generally true. However, to ensure doubly stochastic property, it is possible to design the matrix $W$ through schemes such as the Metropolis-Hastings (M-H) weighting (Schwarz et al., 2014). In M-H weighting schemes, the importance of node $v_i$ for node $v_j$ is inversely proportional to the maximum between the degrees of the two nodes. In particular, for any pair of distinct nodes $(v_i, v_j)$, if there is no edge between $v_i$ and $v_j$ then $w_{ji} = 0$, otherwise $w_{ji} = 1/\max(|\mathcal{E}_i|, |\mathcal{E}_j|)$. Moreover, $w_{ii} = 1 - \sum_{(v_j, w_{ji}) \in \mathcal{E}_i} w_{ji}$.

## 3.2 CONVERGENCE ANALYSIS

In this section, we prove the convergence of the proposed algorithm whenever the matrix $W$ is doubly stochastic and the gradient $\nabla \mathcal{L}(\mathcal{D}_i, \mathcal{X}_i, \Theta_i^t)$ is Lipschitz continuous (Goldstein, 1977). We define the network error $\mathbf{E}$ as the deviation of the node parameters $\Theta_i$ from the network average $\bar{\Theta}$. Particularly, let $\bar{\Theta}^t$ be the averaged parameters of all $\Theta_i^t$ for any $v_i \in \mathcal{V}$. We define the (PIDFL) error at time $t$ as follows:

$$\mathbf{E}_t = \frac{1}{2|\mathcal{V}|} \sum_{i=1}^{|\mathcal{V}|} \left\| \Theta_i^t - \bar{\Theta}^t \right\|^2. \tag{7}$$

The following theorem proves the convergence of Algorithm 1 by showing that the error $\mathbf{E}_t$, decreases over time, that is $\mathbf{E}_{t+1} \leq \beta \mathbf{E}_t$, where $\beta$ is called the convergence rate and $\beta \in [0, 1)$. A smaller $\beta$ implies that the algorithm is reducing the error more rapidly, so the minimum value for $\beta$ is preferred. In the Appendix B, we discuss more on the convergence to an optimal solution and present the generalization bound.

**Theorem 1.** *Let $\langle \mathcal{V}, \mathcal{E}, N, \Delta \rangle$ be a PIDFL, $\langle \mathcal{D}_i, \mathcal{X}_i, \Delta, N, \mathcal{E}_i, \tau, \lambda \rangle$ be an instance of Algorithm 1, and $W$ be the weighted matrix corresponding to weighted graph $\langle \mathcal{V}, \mathcal{E} \rangle$. If the gradient $\nabla \mathcal{L}(\mathcal{D}_i, \mathcal{X}_i, \Theta_i^t)$ is Lipschitz continuous and $W$ is doubly stochastic, then there exists $\beta \in [0, 1)$ such that $\mathbf{E}_{t+1} \leq \beta \mathbf{E}_t$ holds, for any $t \in [0, \tau - 1]$.*

*Proof.* As defined in Eq. (7), we have that:

$\mathbf{E}_t = \frac{1}{2|\mathcal{V}|} \sum_{i=1}^{|\mathcal{V}|} \left\| \Theta_i^t - \bar{\Theta}^t \right\|^2$, and $\mathbf{E}_{t+1} = \frac{1}{2|\mathcal{V}|} \sum_{i=1}^{|\mathcal{V}|} \left\| \Theta_i^{t+1} - \bar{\Theta}^{t+1} \right\|^2$.

Considering Lipschitz continuous conditions for the gradients $\nabla \mathcal{L}(\mathcal{D}_i, \mathcal{X}_i, \Theta_i^t)$, there exists a constant $\kappa$ such that for all $\Theta_r, \Theta_s$, with $r, s \in [1, |\mathcal{V}|]$ we have (Goldstein, 1977):

$$\|\nabla \mathcal{L}(\mathcal{D}_r, \mathcal{X}_r, \Theta_r) - \nabla \mathcal{L}(\mathcal{D}_s, \mathcal{X}_s, \Theta_s)\| \leq \kappa \|\Theta_r - \Theta_s\|.$$

Therefore, in the update step (i.e., Line 7 in Algorithm 1) since matrix $W$ is a doubly stochastic, the average remains the same after combination, that is for any $t < \tau$ we have that:

$$\bar{\Theta}^{t+1} = \frac{1}{|\mathcal{V}|} \sum_{i=1}^{|\mathcal{V}|} \Theta_i^{t+1} = \frac{1}{|\mathcal{V}|} \sum_{i=1}^{|\mathcal{V}|} \sum_{(v_j, w_{ji}) \in \mathcal{E}_i} w_{ji} \Theta_j^t = \frac{1}{|\mathcal{V}|} \sum_{j=1}^{|\mathcal{V}|} (\sum_{(v_i, w_{ji}) \in \mathcal{E}_j} w_{ji}) \Theta_j^t = \bar{\Theta}^t.$$

Consider now the gradient descent update of adaptation (i.e., Line 5 in Algorithm 1), we have that: $\widehat{\Theta}_i^t = \Theta_i^t - \mu \nabla \mathcal{L}(\mathcal{D}_i, \mathcal{X}_i, \Theta_i^t)$. As the gradient is Lipschitz continuous we have that

$$\|\widehat{\Theta}_i^t - \Theta_i^t\| = \mu \|\nabla \mathcal{L}(\mathcal{D}_i, \mathcal{X}_i, \Theta_i^t)\| \leq \mu C \|\Theta_i^t - \bar{\Theta}^t\|$$

where $C$ is the Lipschitz constant. We now expand $\|\Theta_i^{t+1} - \bar{\Theta}^{t+1}\|^2$ as follows:

$$\|\Theta_i^{t+1} - \bar{\Theta}^{t+1}\|^2 = \left\| \sum_{(v_j, w_{ji}) \in \mathcal{E}_i} w_{ji} \widehat{\Theta}_j^t - \bar{\Theta}^t \right\|^2.$$

Applying the convexity of the squared norm (Boyd & Vandenberghe, 2004), we have:

$$\left\| \sum_{(v_j, w_{ji}) \in \mathcal{E}_i} w_{ji} \widehat{\Theta}_j^t - \bar{\Theta}^t \right\|^2 \leq \sum_{(v_j, w_{ji}) \in \mathcal{E}_i} w_{ji} \|\widehat{\Theta}_j^t - \bar{\Theta}^t\|^2.$$

Since it holds that $\|\widehat{\Theta}_j^t - \bar{\Theta}^t\|^2 = \|\Theta_j^t - \mu\nabla\mathcal{L}(\mathcal{D}_j, \mathcal{X}_j, \Theta_j^t) - \bar{\Theta}^t\|^2$, we have that

$$\|\Theta_i^{t+1} - \bar{\Theta}^{t+1}\|^2 \leq \sum_{(v_j, w_{ji}) \in \mathcal{E}_i} w_{ji}\|\Theta_j^t - \mu\nabla\mathcal{L}(\mathcal{D}_j, \mathcal{X}_j, \Theta_j^t) - \bar{\Theta}^t\|^2 \text{ and thus}$$

$$\sum_{i=1}^{|\mathcal{V}|} \|\Theta_i^{t+1} - \bar{\Theta}^{t+1}\|^2 \leq \sum_{i=1}^{|\mathcal{V}|} \sum_{(v_j, w_{ji}) \in \mathcal{E}_i} w_{ji}\|\Theta_j^t - \mu\nabla\mathcal{L}(\mathcal{D}_j, \mathcal{X}_j, \Theta_j^t) - \bar{\Theta}^t\|^2$$

and, as $\sum_{(v_j, w_{ji}) \in \mathcal{E}_i} w_{ji} = 1$, we can rewrite the above inequality as follows:

$$\sum_{i=1}^{|\mathcal{V}|} \|\Theta_i^{t+1} - \bar{\Theta}^{t+1}\|^2 \leq \sum_{i=1}^{|\mathcal{V}|} \underbrace{\|\Theta_i^t - \mu\nabla\mathcal{L}(\mathcal{D}_i, \mathcal{X}_i, \Theta_i^t) - \bar{\Theta}^t\|^2}_{\zeta}$$

We expand $\zeta$ as follows: $\|\Theta_i^t - \bar{\Theta}^t\|^2 - 2\mu\langle\Theta_i^t - \bar{\Theta}^t, \nabla\mathcal{L}(\mathcal{D}_i, \mathcal{X}_i, \Theta_i^t)\rangle + \mu^2\|\nabla\mathcal{L}(\mathcal{D}_i, \mathcal{X}_i, \Theta_i^t)\|^2$ where $\langle \cdot, \cdot \rangle$ represents the inner product. Considering the Lipschitz conditions, we have: $\|\Theta_i^t - \mu\nabla\mathcal{L}(\mathcal{D}_i, \mathcal{X}_i, \Theta_i^t) - \bar{\Theta}^t\|^2 \leq \|\Theta_i^t - \bar{\Theta}^t\|^2(1 - \gamma\mu C + \mathcal{O}(\mu^2 C^2))$ where $\gamma$ is a proportionality constant, and $\mathcal{O}(\mu^2 C^2)$ represents the second-order terms. Therefore, we have $\beta = (1 - \gamma\mu C + \mathcal{O}(\mu^2 C^2))$ and $\sum_{i=1}^{|\mathcal{V}|} \|\Theta_i^{t+1} - \bar{\Theta}^{t+1}\|^2 \leq \beta \sum_{i=1}^{|\mathcal{V}|} \|\Theta_i^t - \bar{\Theta}^t\|^2$, that concludes the proof. $\qquad\square$

## 4 EXPERIMENTAL ANALYSIS

In this section, we discuss the experiment setup and performance evaluation results of the proposed PIDFL architecture. The code and results have been made available online.[4]

### 4.1 EXPERIMENT SETUP

**Dataset.** We consider different physical phenomena with publicly available datasets including the nonlinear Schrödinger (NLS) models that have been utilized to light propagation in optical fibers (Bafghi & Raissi, 2023), air dispersion in the diffusion and transport of pollutants in the atmosphere (Lanser & Verwer, 1999), drug diffusion models (Chasnov, 2019), Burger equation that is used to model fluid dynamics and traffic flow (Rudy et al., 2017), the Schrödinger equation from quantum mechanics (Rudy et al., 2017), and finally, the wave equation models (de Wolff et al., 2021). We compare also the performance of the proposed aggregation algorithm DFLA with the well-known baselines including FedAvg (McMahan et al., 2017b), and SegmentedGossip (Hu et al., 2019), across the mentioned datasets. Comparison of PIDFL with SCAFFOLD(Karimireddy et al., 2020) and DEFDSAM-MGS (Shi et al., 2023) is provided in Appendix.

**Data Distribution.** We consider both IID and non-IID distributions. Non-IID distribution is a practical consideration and arises due to factors such as geographical location, demographics, or device usage patterns (Sánchez Sánchez et al., 2024). For non-IID, we consider *Dirichlet* distribution (with $\alpha$ set to 0.5) to distribute data among the nodes in DFL (Wang et al., 2020; Yurochkin et al., 2019). We also add Gaussian noise to input data with a variance of 0.24. Our initial experiments demonstrated a potential bias with the sorted data. Therefore, we have shuffled the data randomly for a more reliable evaluation. More details on the data distribution are provided in Appendix.

### 4.2 RESULTS

A first question is whether (and how much) the incorporation of domain knowledge into a DFL architecture, thereby resulting in the proposed PIDFL architecture, offers any measurable benefits. To this end, Table 1 (resp., Table 2) reports the results for a setting with fully-connected networks of $n = 10$ (resp., $n = 50$) nodes by varying the regularization parameter $\lambda$ and the dataset—clearly, whenever $\lambda = 0$ we obtain the DFL setting. For each pair of dataset and value of $\lambda$ we report the average test loss of the DFLA algorithm among all nodes, that is $\frac{1}{|\mathcal{V}|}\sum_{v_i \in \mathcal{V}} \mathcal{L}^d(\mathcal{D}_i, \Theta_i)$. [5]

---

[4]Code: https://anonymous.4open.science/r/PIDFL-8EAF/
Results: https://file.io/0M8UM57zGfjO.

[5]We use 80% of the data in each node for training and 20% for the test.

Table 1: Average test loss of the DFLA algorithm for PIDFL architecture ($\lambda \in [0.25, 0.5, 0.75, 1]$)) and DFL architecture (i.e., DFLA with $\lambda = 0$) with $n = 10$ nodes. In cyan we report the gap (in percentage) w.r.t. the case $\lambda = 0$. Bold represents best in row.

| Dataset/PDE | $\lambda = 0$ (DFL) | $\lambda = 0.25$ | $\lambda = 0.5$ | $\lambda = 0.75$ | $\lambda = 1.0$ |
|---|---|---|---|---|---|
| NLS | $.337\pm.051$ | $\mathbf{.110\pm.014}$ | $.124\pm.02$ | $.140\pm.007$ | $.148\pm.004$ |
| | – | **67.434** | 63.082 | 58.411 | 56.168 |
| Air Dispersion | $.190\pm.067$ | $.135\pm.041$ | $\mathbf{.112\pm.025}$ | $.123\pm.038$ | $.147\pm.064$ |
| | – | 29.187 | **41.133** | 35.205 | 22.593 |
| Drug Diffusion | $.087\pm.007$ | $.082\pm.005$ | $.080\pm.012$ | $\mathbf{.077\pm.008}$ | $.092\pm.012$ |
| | – | 6.042 | 7.614 | **11.519** | -5.442 |
| Burger | $.013\pm.005$ | $.009\pm.002$ | $.011\pm.006$ | $\mathbf{.007\pm.002}$ | $.008\pm.002$ |
| | – | 33.925 | 12.659 | **45.572** | 4.154 |
| Schrödinger | $.160\pm.01$ | $.126\pm.015$ | $\mathbf{.108\pm.005}$ | $.126\pm.012$ | $.116\pm.003$ |
| | – | 2.951 | **32.469** | 2.868 | 27.121 |
| Wave | $.028\pm.01278$ | $\mathbf{.027\pm.0055}$ | $.028\pm.0042$ | $.029\pm.0010$ | $.036\pm.0049$ |
| | – | **1.519** | -1.936 | -2.765 | -28.754 |

Table 2: Average test loss of the DFLA algorithm for PIDFL architecture ($\lambda \in [0.25, 0.5, 0.75, 1]$) and DFL architecture (i.e., DFLA with $\lambda = 0$) with $n = 50$ nodes. In cyan we report the gap (in percentage) w.r.t. the case $\lambda = 0$. Bold represents best in row.

| Dataset/PDE | $\lambda = 0$ (DFL) | $\lambda = 0.25$ | $\lambda = 0.5$ | $\lambda = 0.75$ | $\lambda = 1.0$ |
|---|---|---|---|---|---|
| NLS | $.285\pm.028$ | $\mathbf{.187\pm.017}$ | $.218\pm.017$ | $.246\pm.006$ | $.245\pm.020$ |
| | – | **34.153** | 23.455 | 13.471 | 21.151 |
| Air Dispersion | $.140\pm.041$ | $\mathbf{.090\pm.013}$ | $.098\pm.011$ | $.101\pm.019$ | $.094\pm.012$ |
| | – | **36.164** | 3.389 | 28.333 | 33.299 |
| Drug Diffusion | $.083\pm.012$ | $.071\pm.005$ | $.072\pm.003$ | $.070\pm.003$ | $\mathbf{.067\pm.003}$ |
| | – | 14.637 | 13.288 | 14.977 | **18.616** |
| Burger | $.0053\pm.0012$ | $.0035\pm.0007$ | $\mathbf{.0026\pm.0004}$ | $.0045\pm.0004$ | $.0040\pm.0004$ |
| | – | 34.214 | **5.457** | 15.037 | 25.147 |
| Schrödinger | $.0964\pm.0053$ | $.0812\pm.0005$ | $\mathbf{.0788\pm.0001}$ | $.0822\pm.0021$ | $.0806\pm.00003$ |
| | – | 15.796 | **18.193** | 14.756 | 16.399 |
| Wave | $.074\pm.0138$ | $.032\pm.0045$ | $.037\pm.0034$ | $.042\pm.0067$ | $\mathbf{.031\pm.0022}$ |
| | – | 56.757 | 5.000 | 43.243 | **58.108** |

From Tables 1 and 2 we can draw the following conclusions for non-IID distribution. The regularization parameter $\lambda$ affects the performance and the best value (shown in bold) depends on the specific dataset (and thus on the application domain). It is worth noting that, in 21 over the 24 cases (resp., all the cases) of Table 1 (resp., Table 2), the performance of PIDFL is better than the DFL. Moreover, Table 2 shows that when increasing the number of nodes from $n = 10$ to $n = 50$, the PIDFL outperforms the DFL also in the four cases.

Another interesting question is whether the PIDFL architecture continues to offer measurable benefits against other well-known DFL baselines like FedAvg (McMahan et al., 2017b) and SegmentedGossip (SG) (Hu et al., 2019). To this end, Table 3 reports, for each dataset, the average test loss of $i)$ the DFLA algorithm with the best value of $\lambda$ (obtained from Tables 1 and 2) and $ii)$ baselines DFL, FedAvg, and SG. As a result, the PIDFL approach always outperforms all the baselines DFL, FedAvg and SG in both network settings with $n = 10$ and $n = 50$ nodes, and non-IID data distribution. The experiment results with IID data distribution is discussed in the Appendix and supports the superiority of PIDFL.

## 5 RELATED WORK

Recent studies on FL aim to enhance the robustness and performance of both centralized and decentralized settings by optimizing data management and proposing aggregation algorithms (Huang et al., 2024; Xu et al., 2024; Chen et al., 2024). PeFLL is proposed as a personalized FL algorithm that improves accuracy, reduces computation and communication, and offers theoretical guarantees for generalization. PeFLL utilizes a learning-to-learn approach to train an embedding network and a hypernetwork to represent clients in a latent descriptor space (Scott et al., 2024). Despite the advancements in the FL algorithms, the issues caused by the training data are still challenging. Insuf-

Table 3: Average test loss of the DFLA algorithm with $\lambda = 0$ (i.e., DFL), DFLA algorithm with the best value of $\lambda$, and baselines FedAvg and SG with $n = 10$ and $n = 50$ . Bold represents best in row.

| Dataset/PDE | $n = 10$ | | | | $n = 50$ | | | |
|---|---|---|---|---|---|---|---|---|
| | DFLA ($\lambda = 0$) | DFLA (best $\lambda$) | FedAvg | SG | DFLA ($\lambda = 0$) | DFLA(best $\lambda$) | FedAvg | SG |
| NLS | .337 | **.110** | .323 | .317 | .285 | **.187** | .437 | .425 |
| Air Dispersion | .190 | **.112** | .437 | .428 | .140 | **.090** | .527 | .517 |
| Drug Diffusion | .087 | **.077** | 1.700 | 1.544 | .083 | **.067** | 1.926 | 1.753 |
| Burger | .013 | **.007** | .045 | .043 | .0053 | **.0026** | .040 | .037 |
| Schrödinger | .160 | **.108** | .112 | .112 | .0965 | **.0788** | .0788 | .0788 |
| Wave | .028 | **.027** | .593 | .515 | .074 | **.031** | .7665 | .6766 |

ficient data also referred to as "data scarcity", might cause problems in the robustness and accuracy of the DFL models and degrade the performance due to bias or under-fitting (Babbar & Schölkopf, 2019), causing convergence problems (Fahy et al., 2022). To mitigate these issues, several strategies are proposed in the literature. Zhang et al. (Zhang et al., 2023) utilize transfer learning and introduce an FL paradigm for non-intrusive load monitoring at the edge. Generating synthetic data also referred to as data augmentation, is also used to mitigate data scarcity in FL (Goetz & Tewari, 2020; Li et al., 2022). Chen and Vikola study non-IID local data in FL and propose a method to add data made by variational auto-encoders to the local data (Chen & Vikalo, 2023). Hu et al. (Hu et al., 2022) use the synthetic data to train the model instead of the local data. In the literature, synthetic data is mainly utilized to address the communication issue in FL by replacing the large number of parameters in the ML model. PINNs are a class of neural networks that incorporate physics laws, typically in the form of differential equations, into the learning process. They have emerged as a powerful tool, particularly in scientific computing and situations where data is sparse or expensive (Piccialli et al., 2024). The research on PINNs is still ongoing in many fields, including fatigue life prediction (Zhou et al., 2023), solving partial differential equations (PDEs) (Gao et al., 2022), power systems (Huang & Wang, 2022), magnetic image reconstruction (van Herten et al., 2022) and many others (for a recent survey see Wu et al. (2024)). Li et al. (Li et al., 2023) study managing energy across multiple grids and propose a federated multi-agent deep reinforcement learning (DRL) method. They use a physics-informed reward in their proposal. Although the term physics-informed is used, the authors use the physical characteristics of the problem definition and do not mean the PINNs concept. In another study, Chen et al. (Chen et al., 2023) reviewed FL-based X-ray image screening. Instead of sampling the client loss uniformly, they use local messages and physical facts. By being physics-informed, they mean that people have more interest in the images labeled as "HIT" or "MAYBE", which means the substance features being tested are reflected in the images. As the above-mentioned articles (Li et al., 2023; Chen et al., 2023) include the terms "physics-informed" and "federated learning", they do not discuss the idea of using domain knowledge (differentiable equations) in training the ML model of DFL. Therefore, to the best of our knowledge, this is the first time a generalizable architecture is being proposed for DFL.

## 6 CONCLUSIONS AND FUTURE WORK

We proposed PIDFL, a novel decentralized federated learning architecture that incorporates domain knowledge in the form of differential equations. PIDFL improves the learning process by utilizing physics-related PDEs as soft constraints. We also introduced a suitable data-aggregation algorithm (DFLA), proved its convergence, and discussed its computational complexity. We evidenced the efficacy of PIDFL across many datasets, exhibiting substantial performance enhancements in loss reduction relative to conventional decentralized federated learning algorithms like FedAvg and SegmentedGossip. The experimental findings confirm the capability of PIDFL to integrate domain knowledge and learning in decentralized environments, especially in scenarios with non-IID data distributions.

As future work on the proposed framework, we plan to investigate adaptive mechanisms for selecting the regularization parameter $\lambda$, ensuring data fidelity and domain-knowledge adherence across diverse tasks and enhancing the framework's generalizability.

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

APPENDIX

# A  LIMITATIONS

The proposed PIDFL offers an innovative approach for incorporating domain-specific physical knowledge into decentralized learning; however, some limitations needs to be addressed. These limitations do not undermine the merits or applicability of PIDFL but offer insights for future research and enhancement.

As a general limitation on Physics-Informed Neural Networks (PINNs), the effectiveness of the PIDFL framework depends on the precision of the physical model. This constraint is intrinsic to all PINNs and underscores the imperative for careful selection and validation of the domain-specific physical rules $\Delta$ utilized in training. The PIDFL framework is engineered to manage both IID and non-IID data distributions between nodes. Similar to numerous DFL systems, when significant variation occurs in data distribution across nodes, the framework's performance may be affected. In DFL systems, scalability poses a difficulty across extensive networks with numerous nodes. This is because of non-linear relationship between the number of nodes and computational complexity. The PIDFL demonstrates robust performance in experimented mid-sized networks; however, scaling to massive networks may necessitate the utilization of hierarchical topologies to mitigate communication overheads. The efficacy of PIDFL, similarly to existing PINNs, is acutely dependent on the selection of the regularization parameter $\lambda$. Although we have exhibited the framework's robustness across several $\lambda$ values, the ideal selection of this regulization is dependend on the given situation.

# B  CONVERGENCE AND GENERALIZATION BOUND

Here, we discuss the extension of theorem 1 in *strongly convex* and *non-convex* scenarios. We also discuss the generalization bound of the PIDFL.

**Strongly Convex Scenario**  A function $L(\mathcal{D}_i, \mathcal{X}_i, \Theta_i)$ is strongly convex with respect to $\Theta_i$ if there exists a constant $\mu > 0$ such that for any $\Theta_a, \Theta_b$, we have: $L(\mathcal{D}_i, \mathcal{X}_i, \Theta_a) \geq L(\mathcal{D}_i, \mathcal{X}_i, \Theta_b) + \langle \nabla L(\mathcal{D}_i, \mathcal{X}_i, \Theta_b), \Theta_a - \Theta_b \rangle + \frac{\mu}{2} \|\Theta_a - \Theta_b\|^2$. Strong convexity implies that the objective function has a unique global minimizer $\Theta^*$ (Scaman et al., 2017).

In theorem 1, we proved that by the given assumptions, $\mathbf{E}_{t+1} \leq \beta \mathbf{E}_t$. Let each local objective $L(\mathcal{D}_i, \mathcal{X}_i, \Theta_i)$ be strongly convex with a constant $\mu > 0$, therefore, the aggregate objective $L(\Theta) = \frac{1}{|\mathcal{V}|} \sum_{i=1}^{|\mathcal{V}|} L(\mathcal{D}_i, \mathcal{X}_i, \Theta)$ is also strongly convex, with a unique minimizer $\Theta^*$. The contraction of the consensus error, combined with gradient updates, ensures that all nodes' parameters $\Theta_i$ converge to the unique minimizer (Koloskova et al., 2020): $\lim_{t \to \infty} \Theta_i^t = \Theta^*, \quad \forall i \in \mathcal{V}$.

In other words, in theorem 1, having $E_{t+1} \leq \beta E_t$ where $0 \leq \beta < 1$ (by controlling the $\mu$). With $\beta < 1$, we have a decay in $E_t$, therefore: $E_{t+1} \leq \beta E_t \leq \beta^2 E_{t-1} \leq \cdots \leq \beta^{t+1} E_0$, As $t \to \infty$, $\beta^t \to 0$, hence $\lim_{t \to \infty} E_t = 0$. In the strongly convex case, convergence to the unique optimal point $\Theta^*$ follows $\lim_{t \to \infty} \Theta_i^t = \Theta^*, \quad \forall i \in \mathcal{V}$.

**Non-Convex Scenario**  Assuming that the gradients of the local loss functions $\mathcal{L}(\mathcal{D}_i, \mathcal{X}_i, \Theta_i)$ are uniformly bounded (across all nodes and all iterations) and defining $G$ as a bound on the gradient norm of the local loss functions (Arjevani et al., 2023), the global average parameter $\bar{\Theta}^t = \frac{1}{|\mathcal{V}|} \sum_{i \in \mathcal{V}} \Theta_i^t$ converges to a stationary point of the global objective $\mathcal{L}_{\text{global}}(\Theta) = \frac{1}{|\mathcal{V}|} \sum_{i \in \mathcal{V}} \mathcal{L}(\mathcal{D}_i, \mathcal{X}_i, \Theta_i)$ as $t \to \infty$, at the rate of $\mathcal{O}(1/\sqrt{t})$. Assuming the bounded gradient $\|\nabla \mathcal{L}(\mathcal{D}_i, \mathcal{X}_i, \Theta_i)\| \leq G \quad \forall i \in \mathcal{V}$ and since $\mathcal{L}(\cdot)$ is non-convex, the convergence is to a stationary point where $\|\nabla \mathcal{L}_{\text{global}}(\Theta)\| = 0$.

To prove, we define $e^t = \Theta_i^t - \bar{\Theta}^t$ as the deviation of local parameters from the global average. Substituting the update rule (Algorithm 1) into the deviation, we have: $\mathbf{e}^{t+1} = W \mathbf{e}^t$, where $\mathbf{e}^t = [e_1^t, e_2^t, \ldots, e_{\mathcal{V}}^t]^\top$. Considering the doubly stochastic $W$, the global average $\bar{\Theta}^t$ is invariant, so the deviation $\mathbf{e}^t$ evolves independently. Therefore: $\|\mathbf{e}^{t+1}\| = \|W \mathbf{e}^t\| \leq \lambda_2(W) \|\mathbf{e}^t\|$. By iterating this over $t$ steps, we have: $\|\mathbf{e}^t\| \leq \lambda_2(W)^t \|\mathbf{e}^0\|$, that indicates a geometric decay of $\|\mathbf{e}^t\|$ at a rate proportional to $\lambda_2(W)$. Since $\lambda_2(W) < 1$, it follows that $\|\mathbf{e}^t\| \to 0$ as $t \to \infty$. This implies that

all local parameters $\Theta_i^t$ converge to the global average $\bar{\Theta}^t$. Therefore, $\|e^t\|^2 \leq \lambda_2(W)^t \|e^0\|^2$. This indicates that the consensus error diminishes geometrically over iterations, i.e., $\|e^t\| \rightarrow 0$ as $t \rightarrow \infty$. Therefore, bounding the gradient norm as $\|\nabla \mathcal{L}_{\text{global}}(\Theta)\| \leq \frac{1}{|\mathcal{V}|} \sum_{i \in \mathcal{V}} \|\nabla \mathcal{L}(\mathcal{D}_i, \mathcal{X}_i, \Theta_i)\| + \|e^t\|$ and using the diminishing consensus error $\|e^t\| \rightarrow 0$ and bounded gradients $\|\nabla \mathcal{L}(\cdot)\| \leq G$, the global gradient norm satisfies $\mathbb{E}[\|\nabla \mathcal{L}_{\text{global}}(\Theta^t)\|^2] \leq \mathcal{O}(1/\sqrt{t})$.

**Generalization Bound**  For generalization bound (Bousquet & Elisseeff, 2002), we assume that the local loss function $L(\mathcal{D}_i, \mathcal{X}_i, \Theta)$ is $C$-smooth (Lipschitz continuous with constant $C$), and $\gamma = \lambda_2(W) - \lambda_1(W) > 0$ represents the eigengap of a doubly stochastic weighing matrix $W$, where $\lambda_1(W) = 1$ is the largest eigenvalue and $\lambda_2(W)$ is the second-largest eigenvalue. The data heterogeneity across nodes is bounded by $\delta$. The $\delta$ measures the variation in data distributions across nodes and can be quantified by the difference between local and global empirical risks: $\delta = \frac{1}{|\mathcal{V}|} \sum_{i=1}^{|\mathcal{V}|} \left| \hat{R}_i(\Theta) - \hat{R}(\Theta) \right|$, where $\hat{R}_i(\Theta)$ and $\hat{R}(\Theta)$ are the local and global empirical risks, respectively (Bousquet & Elisseeff, 2002). We define the generalization bound as $E_{gen} = \mathbb{E}\left[ R(\Theta) - \hat{R}(\Theta) \right]$ (Bousquet & Elisseeff, 2002). This definition shows the difference between the model's expected performance on unseen data and its performance on the training data. Let $R(\Theta) = \mathbb{E}_{(\mathcal{D}, \mathcal{X})}[L(\mathcal{D}, \mathcal{X}, \Theta)]$ be the expected global risk, and $\hat{R}(\Theta) = \frac{1}{|\mathcal{V}|} \sum_{i=1}^{|\mathcal{V}|} \hat{R}_i(\Theta_i)$ be the global empirical risk, with $\hat{R}_i(\Theta_i) = \frac{1}{|\mathcal{D}_i|} \sum_{\mathcal{D}_i} L(\mathcal{D}_i, \mathcal{X}_i, \Theta_i)$. Therefore, $E_{gen} = \mathbb{E}\left[ R(\Theta) - \hat{R}(\Theta) \right] = \mathbb{E}\left[ R(\Theta) - \frac{1}{|\mathcal{V}|} \sum_{i=1}^{|\mathcal{V}|} R_i(\Theta_i) \right] + \mathbb{E}\left[ \frac{1}{|\mathcal{V}|} \sum_{i=1}^{|\mathcal{V}|} \left( R_i(\Theta_i) - \hat{R}_i(\Theta_i) \right) \right]$.

The first term represents the error due to the lack of compatibility between the local models $\Theta_i$ and the global model $\Theta$ (the $\Theta$ after convergence), while the second term captures the error due to the empirical approximation at each node. Using the Lipschitz continuity of the gradient with constant $C$, and applying $\|R(\Theta) - R_i(\Theta_i)\| \leq C\|\Theta_i - \Theta\|$, considering the doubly stochastic property of the weight matrix $W$ we have: $\mathbb{E}\left[ R(\Theta) - \frac{1}{|\mathcal{V}|} \sum_{i=1}^{|\mathcal{V}|} R_i(\Theta_i) \right] \leq \frac{C}{|\mathcal{V}|} \sum_{i=1}^{|\mathcal{V}|} \|\Theta_i - \Theta\|$. The eigengap bounds the rate of consensus between nodes, i.e. $\|\Theta_i - \Theta\|^2 \leq \frac{1}{\sqrt{\gamma}} \|\Theta_i - \Theta\|$. Therefore, $\mathbb{E}\left[ R(\Theta) - \frac{1}{|\mathcal{V}|} \sum_{i=1}^{|\mathcal{V}|} R_i(\Theta_i) \right] \leq \frac{C}{\sqrt{\gamma}} \cdot \frac{1}{|\mathcal{V}|} \sum_{i=1}^{|\mathcal{V}|} \|\Theta_i - \Theta\|^2$. The second term as $\mathbb{E}\left[ \frac{1}{|\mathcal{V}|} \sum_{i=1}^{|\mathcal{V}|} \left( R_i(\Theta_i) - \hat{R}_i(\Theta_i) \right) \right]$ is bounded by a constant $\delta$, which depends on the size of the local dataset $|\mathcal{D}_i|$ and the heterogeneity of data across nodes. Formally speaking, $\mathbb{E}\left[ \frac{1}{|\mathcal{V}|} \sum_{i=1}^{|\mathcal{V}|} \left( R_i(\Theta_i) - \hat{R}_i(\Theta_i) \right) \right] \leq \delta$. Combining the bounds for the two terms, we have: $E_{gen} = \mathbb{E}\left[ R(\Theta) - \hat{R}(\Theta) \right] \leq \frac{C}{\sqrt{\gamma}} \cdot \frac{1}{|\mathcal{V}|} \sum_{i=1}^{|\mathcal{V}|} \|\Theta_i - \Theta\|^2 + \delta$. The first term, $\frac{C}{\sqrt{\gamma}} \cdot \frac{1}{|\mathcal{V}|} \sum_{i=1}^{|\mathcal{V}|} \|\Theta_i - \Theta\|^2$, presents the impact of network connectivity ($\gamma$) and the smoothness of the loss function ($C$) on the generalization error. The second term, $\delta$, accounts for the approximation error due to finite data and heterogeneity across nodes. In our experiments, we utilized *Dirichlet distribution* with parameter $\alpha$ to model the heterogeneity of the data. Larger values of $\alpha$ result in distributions closer to IID. Empirical analysis (like what is performed by (Hsu et al., 2019)) suggests that $\delta$ is inversely related to $\alpha$, approximately following $\delta \propto \frac{1}{\alpha}$.

**Discussion on Convergence Rate**  The convergence rate $\mathcal{O}(\mu^2 C^2)$ reflects the influence of two critical factors the step size $\mu$ (learning rate) and the Lipschitz constant $C$, which bounds the gradient of the loss function $\mathcal{L}$. The Lipschitz constant $C$ represents the smoothness of the loss function $\mathcal{L}$. A smaller $C$ implies that the loss surface is smoother, which can lead to more stable and efficient optimization. The quadratic dependence $C^2$ indicates that a higher Lipschitz constant (i.e., less smoothness) slows down the convergence process. This is because larger $C$ leads to greater variability in the gradients, which the algorithm must account for by taking smaller steps. The convergence rate $\mathcal{O}(\mu^2 C^2)$ ensures that the algorithm effectively handles smooth loss functions by leveraging $C$ as a control measure for gradient variations. When $C$ is small, the optimization benefits from faster convergence, making the algorithm suitable for problems with well-behaved loss functions. For functions with high Lipschitz constants (large $C$), the quadratic dependence highlights the sensitivity of the algorithm to the smoothness of $\mathcal{L}$. In cases with such challenges, techniques such as gradient clipping or adaptive learning rates may be employed to mitigate the adverse effects. The dependence on $\mu^2$ implies that the step size (learning rate) must be carefully chosen. A smaller $\mu$

leads to slower convergence, while a larger $\mu$ could exacerbate the impact of $C$, causing instability. For real-world problems, the value of $C$ can often be estimated or bounded based on empirical observations of the loss landscape. For instance, in decentralized learning scenarios, smoother functions (smaller $C$) may arise naturally from averaging techniques. While a squared convergence rate may not be optimal, it is still significant in non-convex settings as it guarantees the algorithm will approach a stationary point. In distributed and federated learning frameworks, where non-convex loss surfaces are common, achieving even $\mathcal{O}(\mu^2 C^2)$ convergence provides a reliable method for gradual improvement, especially given the challenges posed by communication constraints and heterogeneous data. For faster convergence, we may increase $\mu$, but this could lead to larger consensus errors or divergence if $\mu$ exceeds stability bounds. Choosing an optimal $\mu$ balances convergence speed with stability, particularly important in decentralized settings (Wang et al., 2019; Arjevani et al., 2023).

## C    EXPERIMENTAL INSIGHTS

The experiments were performed on a machine featuring a 2.93 GHz base processor speed, 12 physical cores, 24 logical processors, and 64 GB of RAM. We report the average loss on the test set after, that is $\frac{1}{|\mathcal{V}|} \sum_{v_i \in \mathcal{V}} \mathcal{L}^{\mathrm{d}}(\mathcal{D}_i, \Theta_i)$. Table 4 presents the parameters used in our performed experiments which were selected based on common assumptions or practical evaluations.

Table 4: Settings used in our performed experiments.

| Parameter | Value |
|---|---|
| Network Size (Nodes) | 10 and 50 |
| Iterations (communication rounds) | 100 |
| Non-IID Data Distribution | Dirichlet distribution with $\alpha = 0.5$ |
| Learning Rate | 0.1 |
| Noise | Gaussian with variance 0.24 |

**Impact of Communication Rounds**    A critical determinant affecting the efficacy of the PIDFL framework is the number of iterations $\tau$ (or Communication Rounds, CRs) in Algorithm 1. The impacts of $\tau$ on the experiments are presented in the following.

In experiments with 100 iterations, the models demonstrated a more accelerated decrease in mean loss relative to scenarios with fewer CRs. This is especially evident in the outcomes for the *NLS*, *Air Dispersion*, and *Burger* datasets. This suggests that increased $\tau$, enables decentralized nodes to more effectively synchronize their model parameters, hence enhancing overall performance.

Although increasing the number of iterations may enhance convergence, it simultaneously results in greater communication overhead. In massive networks, this may result in considerable delays due to large amount of data transmission across nodes. For instance, in the experiments with $n = 50$ nodes, the communication overhead has become increasingly evident. Although the average loss consistently diminished with additional communication cycles, the enhancement was less significant relative to the $n = 10$ node networks. This indicates that although communication aids in aligning model parameters, there might be a diminishing return with increased communication cycles in larger networks, particularly when accounting for the associated computational and temporal costs.

When the data distribution among nodes is non-IID, increasing $\tau$ facilitates greater information sharing among nodes, hence diminishing the discrepancies in their local models. Experimental results indicate that models with higher number of iterations, show enhanced performance in non-IID data distributions. This is more apparent in the *Drug Diffusion* and *Burger* datasets. Figure 3 illustrates the test loss value across communication rounds for Air Dispersion and Wave Datasets.

In decentralized federated learning settings with peer to peer communications there is a need to frequent communication between nodes to ensure consistency across local models. The use of a doubly stochastic weight matrix $W$ enables the proposed PIDFL to achieve geometric convergence of the consensus error. While this may increase communication overhead, it is balanced by the

benefit of rapid and robust convergence to a consensus model. In practical implementations, the communication frequency can be reduced by employing asynchronous communication or periodic synchronization intervals. Techniques such as model compression can mitigate the communication overhead without significantly affecting performance. Although our experiment results focus on fully connected networks, the proposed architecture and aggregation is compatible with partially connected topologies. The doubly stochastic weight matrix $W$ can be constructed for partially connected networks using techniques such as the Metropolis-Hastings weighting scheme or graph Laplacian-based methods. These approaches ensure that the necessary mathematical properties for convergence, such as bounded spectral gaps, are preserved even when the communication graph is not fully connected.Considering fully connected networks we put emphasize the impact of incorporating domain knowledge through the regularization term $\lambda \mathcal{L}_\Delta$. By reducing the variability introduced by different communication topologies, we could more clearly demonstrate the effectiveness of domain-specific information in improving performance. This design choice enables a more controlled and interpretable analysis of the contributions of our method.

**IID Data Distribution** In Table 5 we report the average test loss of the DFLA algorithm for PIDFL architecture ($\lambda \in [0.25, 0.5, 0.75, 1]$) and DFL architecture ($\lambda = 0$) with $n = 10$ nodes and IID data distribution. It is worth noting that the performance of PIDFL is better than the DFL in all datasets and regulization parameters ($\lambda \in [0.25, 0.5, 0.75, 1]$). Moreover, Table 6 reports, for each dataset, the average test loss of $i$) the DFLA algorithm with the best value of $\lambda$ (obtained from Tables 5 ) and $ii$) baselines DFLA, FedAvg, and SG when the data distribution is IID. As shown, the proposed approach outperforms all the baselines.

Table 5: Average test loss of the DFLA algorithm for PIDFL architecture ($\lambda \in [0.25, 0.5, 0.75, 1]$) and DFL architecture ($\lambda = 0$) with $n = 10$ nodes and IID data distribution. In cyan we report the gap (in percentage) w.r.t. the case $\lambda = 0$. Bold represents best in row.

| Dataset/PDE | $\lambda = 0$ (DFL) | $\lambda = 0.25$ | $\lambda = 0.5$ | $\lambda = 0.75$ | $\lambda = 1.0$ |
|---|---|---|---|---|---|
| NLS | .284±.026 | **.153±.019** | .174±.014 | .170±.023 | .185±.010 |
| | – | 46.186 | 38.556 | 4.234 | 34.777 |
| Air Dispersion | .177±.0409 | .092±.0052 | .128±.0451 | .154±.0543 | **.078±.0023** |
| | – | 48.045 | 27.650 | 13.010 | 56.106 |
| Drug Diffusion | .085±.014 | .062±.002 | .068±.005 | **.058±.0002** | .063±.004 |
| | – | 27.208 | 19.566 | 31.437 | 25.916 |
| Burger | .011±.00001 | .008±.004 | .007±.003 | **.006±.002** | .007±.001 |
| | – | 2.767 | 32.386 | 41.001 | 32.015 |
| Schrödinger | .147±.017 | .104±.0003 | .096±.0005 | .094±.0033 | **.092±.0019** |
| | – | 29.642 | 34.663 | 35.930 | 37.782 |
| Wave | .044±.024 | **.018±.002** | .024±.006 | .019±.002 | .019±.0004 |
| | – | 59.892 | 44.387 | 56.323 | 55.816 |

**Non-IID Data Distribution** In DFL, data is frequently distributed in a non-IID fashion. In contrast to IID environments, where each node accesses analogous data distributions, non-IID distributions more accurately reflect real-world situations in which the nodes produce data locally, resulting in diversity in data distributions among nodes. Similar to many DFL studies, we employ the *Dirichlet* distribution for non-IID data distributions (Shi et al., 2023). As shown below, the Dirichlet distribution is defined by a concentration parameter $\alpha$ that governs the heterogeneity of the data distribution, and is defined as follows:

$$P(\mathbf{p}_i \mid \alpha) = \frac{1}{B(\alpha)} \prod_{i=1}^{K} p_i^{\alpha_i - 1},$$

where $K$ denotes the number of classes, $B(\alpha)$ represents the beta function to normalize the distribution, and $\mathbf{p}_i$ represents the probability vector for the $i$-th class. The In our experiments, we utilized Dirichlet distribution with $\alpha = 0.5$ to allocate the data across $N$ nodes. Figure 4 shows both IID and non-IID distributions over NLS dataset.

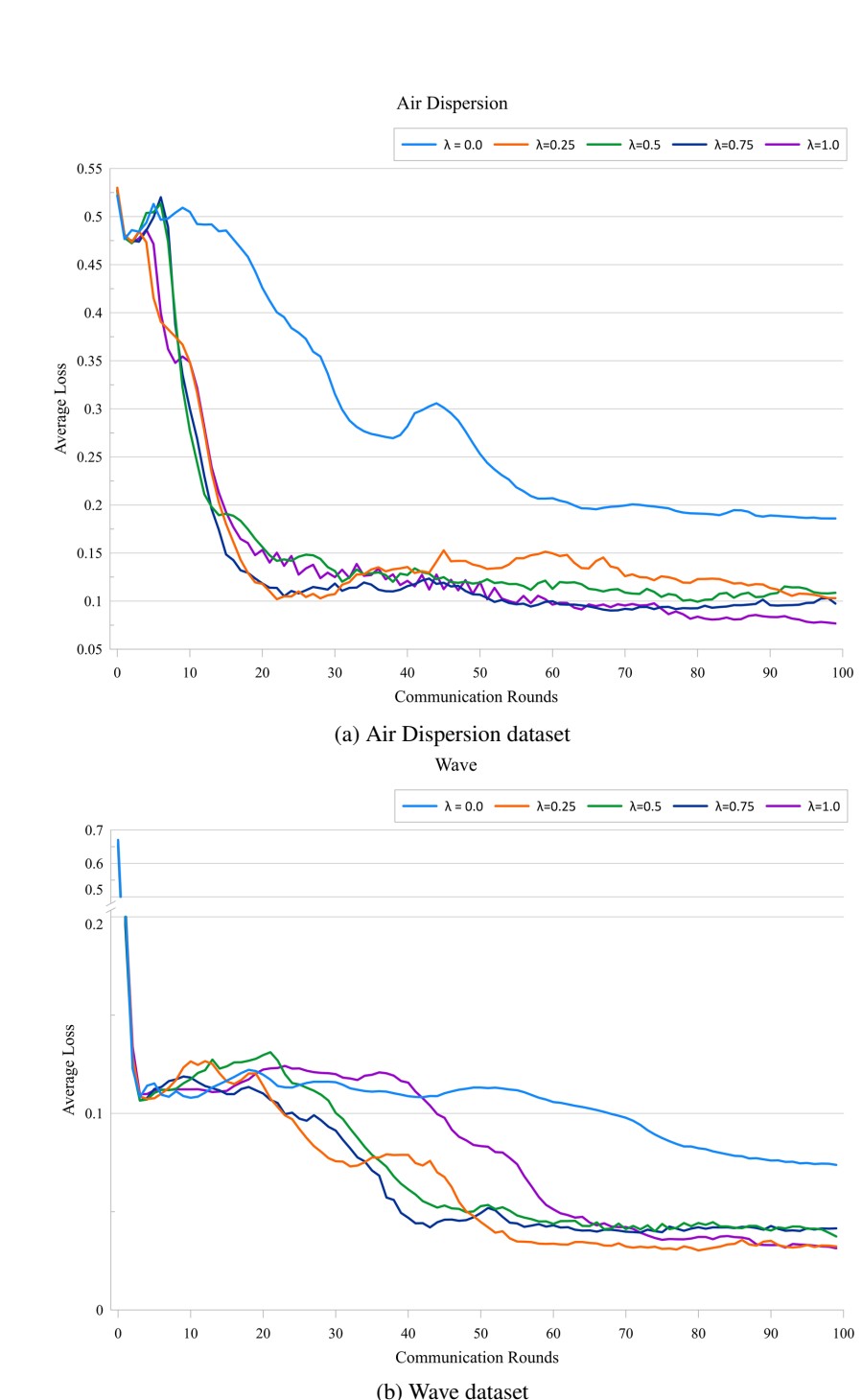

(a) Air Dispersion dataset

(b) Wave dataset

Figure 3: Average Loss over communication rounds for different values of regularization parameter $\lambda$, where blue ($\lambda = 0$), orange ($\lambda = 0.25$), green ($\lambda = 0.5$), dark-blue ($\lambda = 0.75$), and purple ($\lambda = 1$). These plots show how loss decreases as communication rounds increase for both the Air Dispersion and Wave datasets.

Table 6: Comparison of the proposed PIDFL's average loss with FedAvg and Segmented Gossip (SG) techniques on IID data for 10 nodes

| Dataset/PDE | 10 Nodes | | | |
|---|---|---|---|---|
| | DFL ($\lambda = 0$) | DFLA (best $\lambda$) | FedAvg | SG |
| NLS | .284 | **.153** | .364 | .356 |
| Air Dispersion | .177 | **.078** | .348 | .337 |
| Drug Diffusion | .085 | **.058** | 1.820 | 1.653 |
| Burger | .011 | **.006** | .024 | .023 |
| Schrödinger | .147 | **.092** | .091 | .090 |
| Wave | .044 | **.018** | .739 | .653 |

**Comparing PIDFL with SCAFFOLD and DFedSAM-MGS** We mainly consider the *FedAvg* and *Segmented Gossip* to compare the performance of PIDFL, since these methods are well-known as the baselines in DFL. However, to have a better understanding, we also compare the PIDFL with some of more recent methods in the literature including SCAFFOLD (Karimireddy et al., 2020) and DEFDSAM-MGS (Shi et al., 2023) on both IID and non-IID settings. The results are privided in Table 7 show that the proposed PIDFL outperforms the existing DFL algorithms because of its inherent strength that comes from domain knowledge. These results indicate that although the aggregation algorithms play a key role, especially in controlling the communication overheads in DFL, the integration of domain knowledge can provide more effect on the accuracy. The results also indicate that for non-IID settings, the PIDFL shows significant improvement, particularly in complex datasets like "Air Dispersion" and "Wave."

Table 7: Comparison of the proposed PIDFL's average loss with FedAvg, Segmented Gossip (SG), SCAFFOLD, DEFDSAM-MGS techniques on IID and non-IID data for 10 nodes

| Dataset/PDE | 10 Nodes (Non-IID) | | | | | 10 Nodes (IID) | | | | |
|---|---|---|---|---|---|---|---|---|---|---|
| | PIDFL | FedAvg | SG | SCAFFOLD | DFEDSAM | PIDFL | FedAvg | SG | SCAFFOLD | DFEDSAM |
| NLS | .110 | .323 | .317 | .398 | .484 | .153 | .364 | .356 | 0.380 | .387 |
| Air Dispersion | .112 | .437 | .428 | .495 | .444 | .078 | .348 | .337 | 0.354 | .407 |
| Drug Diffusion | .077 | 1.700 | 1.544 | .072 | .068 | .058 | 1.820 | 1.653 | 0.063 | .062 |
| Wave | .027 | .593 | .515 | .123 | .117 | .018 | .739 | .653 | 0.106 | .108 |

**Choosing the Subset $\mathcal{X}$** The subset $\mathcal{X}_i \subset D_i$ can be chosen adaptively based on the relevance of the domain knowledge to the data. This adaptive selection process could be designed to prioritize data points where the PDE constraints are expected to have the most significant impact on model accuracy (depending on the application). For instance, regions in the dataset associated with higher physical variations or boundary conditions may be selected to ensure that the model learns crucial domain-specific behaviors. While expert knowledge can guide the initial selection criteria, the process is further refined through iterative model training, where the contribution of each data point to the PDE constraint is evaluated. This iterative refinement enables the model to self-adjust its focus on parts of the dataset $\mathcal{X}_i$ where domain knowledge is most informative, reducing dependence on manual selection by experts. In case none of these options are available, the system can select a subset of the $D_i$ randomly to present the $\mathcal{X}_i$.

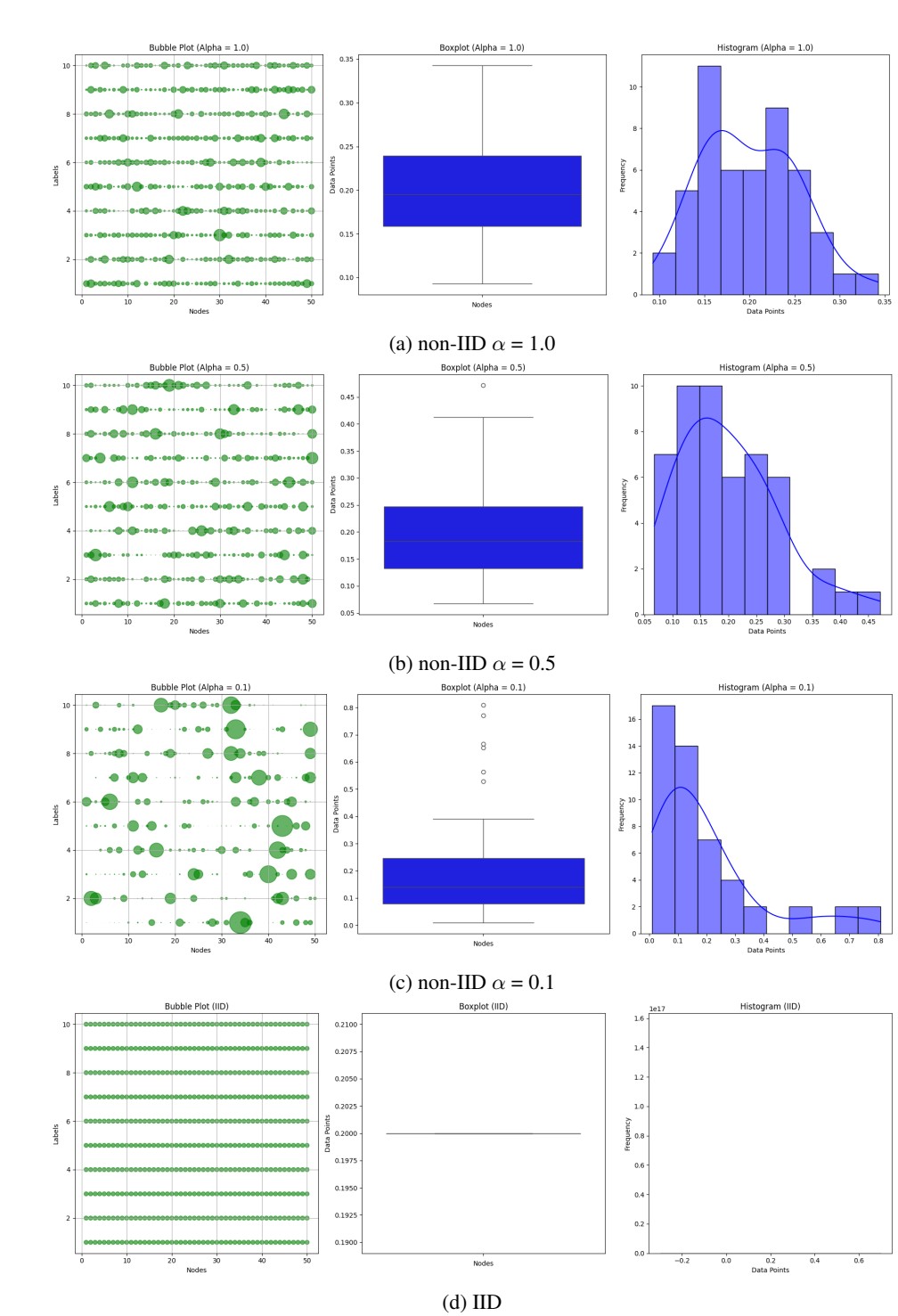

(a) non-IID $\alpha = 1.0$

(b) non-IID $\alpha = 0.5$

(c) non-IID $\alpha = 0.1$

(d) IID

Figure 4: Data distribution plots for the NLS dataset for non-IID (a, b and c) and IID distributions (d). For non-IID, the Dirichlet distribution is plotted with $\alpha \in \{0.1, 0.5, 1\}$. Larger bubbles represent more data assigned to a node. The plots demonstrate the distribution of data across nodes for different levels of non-IID-ness (as controlled by the Dirichlet $\alpha$ values) and a fully IID distribution.

