# OpenReview forum: "Physics-Informed Decentralized Federated Learning"
_ICLR.cc/2025/Conference — ICLR 2025 Conference Withdrawn Submission_

### Official Review · Reviewer_odc6 · 2024-10-30

**Soundness:** 2
**Presentation:** 2
**Contribution:** 3
**Rating:** 3
**Confidence:** 4

**Summary:**

This paper proposes a federated learning algorithm with theoretical guarantees by integrating physics-informed neural networks with decentralized federated learning. Leveraging domain knowledge, this algorithm demonstrates superior performance on non-IID datasets.

**Strengths:**

This paper proposes a federated learning algorithm with theoretical guarantees by integrating physics-informed neural networks with decentralized federated learning. Leveraging domain knowledge, this algorithm demonstrates superior performance on non-IID datasets.

**Weaknesses:**

The role of domain knowledge is insufficiently clarified—is it incorporated in the form of partial differential equations (PDEs) or the choice of dataset $\mathcal{X}$? Furthermore, the PDEs used for training are not adequately specified, nor is there an explanation of how the dataset $\mathcal{X}$ is selected adaptively without expert knowledge.

**Questions:**

1. Is the proposed algorithm synchronous? If so, what impact does idle time in P2P communication have on the algorithm’s performance? Additionally, what is the extra communication overhead introduced compared to a centralized algorithm? The authors are encouraged to analyze this aspect.
2. The incorporation of domain knowledge is insufficiently elaborated. In which aspects is it utilized? A summary dedicated to this would be beneficial.
3. There are errors in the theoretical proofs. In the penultimate equation on line 362, summing over $i$ includes both $w_{ij}$ and $\Theta$, meaning $i$ and $w_{ij}$ cannot be grouped in parentheses independently. Additionally, the result on line 367 does not lead to the result on line 394, as $\gamma$ does not exist. The authors need to thoroughly verify their proofs.
4. The algorithms selected for comparison in the simulations are more than five years old, which might be outdated. Additionally, why was a noise variance of exactly 0.24 chosen—is there any specific reasoning behind this particular value? The authors should provide an explanation.

---

### Official Review · Reviewer_f2zS · 2024-11-04

**Soundness:** 3
**Presentation:** 3
**Contribution:** 3
**Rating:** 6
**Confidence:** 3

**Summary:**

The paper proposes a Physics-Informed Decentralized Federated Learning (PIDFL) framework that incorporates domain-specific knowledge, represented by differential equations, into decentralized federated learning (DFL).

**Strengths:**

- This paper is generally well-written and easy to follow
- Incorporating domain knowledge through physics-informed constraints is novel and it seems to be enhancing model accuracy.
- This paper provides theoretical convergence proof for the DFLA algorithm.
- Experimental results indicate that PIDFL consistently outperforms traditional DFL approaches.

**Weaknesses:**

- It seems that the effectiveness of physics-informed constraints is contingent on the accuracy of the domain knowledge.
- The sensitivity of the regularization parameter is not fully discussed.
- The convergence is at $O(\mu^2C^2)$, what does it say? is squared convergence good? more explanations, insights and remarks on this would be beneficial to this paper
- Many other FL limitations are not discussed, e.g. it seems that this method requires frequent peer-to-peer communication, partial participation is not considered?

**Questions:**

see weakness

---

### Official Review · Reviewer_AApY · 2024-11-04

**Soundness:** 2
**Presentation:** 2
**Contribution:** 2
**Rating:** 3
**Confidence:** 4

**Summary:**

This paper studies integrating domain knowledge into decentralized federated learning (DFL). Specifically, the paper considers local clients training with the Physics-Informed Neural Networks (PINNs) and proposes the PIDFL architecture. Extensive experiments verify the feasibility of the proposed method.

**Strengths:**

+ It is somewhat new to apply the PINNs into DFL.
+ Convergence analysis is provided.

**Weaknesses:**

- This is just a direct and simple application of the PINNs, where no technical challenges are brought up by combining DFL and PINNs.
- The proof is problematic. Why the contraction of consensus error represents the convergence of the algorithm? Should we further prove $\bar{\theta}\rightarrow \theta^{*}$ for the strongly convex case or $|\Vert \nabla f(\theta)\Vert^{2}|\rightarrow 0$ for the non-convex case?
- More baselines should be compared, such as Scaffold, which has better performance than FedAvg.

**Questions:**

See the weakness section.

---

### Official Review · Reviewer_XbvZ · 2024-11-12

**Soundness:** 2
**Presentation:** 3
**Contribution:** 2
**Rating:** 5
**Confidence:** 4

**Summary:**

This paper proposes to leverage physical information into federated learning. By incorporating the corresponding differential equations into the model as penalty term, this paper proposes an algorithm for decentralized federated learning setting.

**Strengths:**

- The writing is easy to understand.

- The idea is intuitive and the algorithm is easy to implement.

**Weaknesses:**

- Lack of motivation: this paper only provided 1-2 examples to motivate the problem settings. I am not convinced that the proposed framework could be used for many applications.

- Lack of theoretical support: no generalization bound

- The convergence results don't seem to be right. As it only says that it will converge to the average of its neighbors weights, it does not imply that it will converge to the (local) optimal solution.

- The idea is simple and it doesn't have much technical contribtuion.

**Questions:**

- How could (1) be written in the form of one in line 138?

- Equation (4). Could we explicitly define "residual" function?

- Line 322. Is this scalable?

- Theorem 1. What does the solution converge to?

---

### Note · Authors · 2024-11-26

**Comment:**

Thank you for taking the time to review our paper. We found several of comments valuable in enhancing the quality of this research.

**Withdrawal Confirmation:**

I have read and agree with the venue's withdrawal policy on behalf of myself and my co-authors.